

# FedDL: personalized federated deep learning for enhanced detection and classification of diabetic retinopathy

Dasari Bhulakshmi and Dharmendra Singh Rajput

School of Computer Science Engineering and Information Systems, Vellore Institute of Technology, Vellore, Tamil Nadu, India

## ABSTRACT

Diabetic retinopathy (DR) is a condition that can lead to vision loss or blindness and is an unavoidable consequence of diabetes. Regular eye examinations are essential to maintaining a healthy retina and avoiding eye damage. In developing countries with a shortage of ophthalmologists, it is important to find an easier way to assess fundus photographs taken by different optometrists. Manual grading of DR is time-consuming and prone to human error. It is also crucial to securely exchange patients' fundus image data with hospitals worldwide while maintaining confidentiality in real time. Deep learning (DL) techniques can enhance the accuracy of diagnosing DR. Our primary goal is to develop a system that can monitor various medical facilities while ensuring privacy during the training of DL models. This is made possible through federated learning (FL), which allows for the sharing of parameters instead of actual data, employing a decentralized training approach. We are proposing federated deep learning (FedDL) in FL, a research paradigm that allows for collective training of DL models without exposing clinical information. In this study, we examined five important models within the FL framework, effectively distinguishing between DR stages with the following accuracy rates: 94.66%, 82.07%, 92.19%, 80.02%, and 91.81%. Our study involved five clients, each contributing unique fundus images sourced from publicly available databases, including the Indian Diabetic Retinopathy Image Dataset (IDRiD). To ensure generalization, we used the Structured Analysis of the Retina (STARE) dataset to train the ResNet50 model in a decentralized learning environment in FL. The results indicate that implementing these algorithms in an FL environment significantly enhances privacy and performance compared to conventional centralized learning methods.

# INTRODUCTION

When diabetic retinopathy (DR) is not identified in its early stages, it can become the primary cause of blindness among working-age adults globally. This eye disease damages the blood vessels in the retina as a result of high blood sugar levels. The damaged vessels can leak, causing blurry vision or even complete vision loss. Individuals with long-standing diabetes are especially vulnerable, particularly if they do not receive proper diagnosis and

Corresponding author
Dharmendra Singh Rajput,
dharmendrasingh@vit.ac.in

treatment. Early detection of the disease allows for effective medical treatment (*Mohan et al., 2023*). Estimates suggest that the number of people affected by DR could reach 700 million by 2045. Consequently, individuals with diabetes need to have regular eye check-ups to prevent vision loss. As the severity of DR increases, the risk of visual impairment also rises (*Ruamviboonsuk et al., 2022*; *Yadav et al., 2023*). Deep learning (DL) has been employed to detect and classify various eye conditions using retinal photos, optical coherence tomography (OCT), and OCT-angiography images. The training process for DL often requires large and diverse datasets to be gathered and stored in a "centralized location" to ensure the model performs well in various situations. However, this method of sharing data can pose real challenges for patient privacy and data security (*Basha et al., 2017*). The study presents a framework that utilizes machine learning (ML) classifiers to diagnose diabetes. The classifiers used include logistic regression, support vector machines (SVM), random forest, decision tree, naive Bayes, and K-nearest neighbor. The SVM classifier is particularly notable for achieving an accuracy of 96.0%. The model aims to increase awareness of diabetes in rural areas and enhance treatment methods (*Rajput et al., 2022*). Federated learning (FL) is a machine learning (ML) approach that involves training the model across multiple decentralized edge devices or servers, each of which stores local data. FL allows the model to be trained on data distributed across many sites without the need to exchange raw data, as opposed to collecting all the data in one central location (*Sharma & Kumar, 2023*). The collaborative model training method allows for multiple contributors to work together on model training while ensuring privacy and data security. FL, a distributed collaborative learning model, enables coordination among several contributors without the need to share sensitive information. This distributed training strategy significantly reduces the risk of data leakage from data sharing or centralization, and ensures data privacy among many organizations (*Rauniyar et al., 2023*).

## Motivation

The motivation behind this study is to address gaps in DR detection by leveraging federated learning. This novel approach allows training machine learning models across decentralized devices without exchanging local data, enhancing privacy and utilizing diverse datasets. DR, a major diabetes complication, can lead to blindness if not detected early. Current methods face inefficiencies, inconsistencies, and limited accessibility, especially in remote areas. Federated learning offers a promising solution by developing accurate and scalable DR detection models, facilitating more efficient screening programs without compromising patient privacy.

## Contributions

The major contributions of this article are as follows:

- We present a new FedDL model for the central server paradigm using FL, which significantly improves the accuracy of DR stage classification.
- Data augmentation was performed on the fundus photographs.

- Five cutting-edge CNN architectures, including ResNet50, DenseNet201, AlexNet, EfficientNetB7, and VGGNet19, were used in the FL environment.
- The experimental evaluations indicate that the proposed methodology exceeds the performance of traditional FL approaches. Specifically, the FedDL method for training DL models in a distributed environment demonstrates enhanced accuracy.

### Article organization

The article is organized into several sections. "Background and Related Work" provides an overview of background and related work in DL and FL. "Proposed Methodology" describes the proposed framework, including database preparation, pre-processing, FL approaches, and DR classification accuracy improvement. "Results and Discussion" presents the results and discussion, including experimental setup, and result analysis. "Conclusion and Future Directions" provides the conclusion and future research directions.

## BACKGROUND AND RELATED WORK

Diabetic retinopathy is a leading cause of vision impairment and blindness in people with long-standing diabetes. Accurate detection of DR is crucial for ophthalmologists, as it can greatly help them manage the increasing number of patients and their corresponding medical records. DL is a crucial approach for analyzing and testing medical images for multiclass classification, segmentation, localization, *etc*. It can quickly learn from and accurately identify images to support human specialists with quantitative outcomes. Convolutional neural networks (CNNs) have demonstrated robust performance in image categorization tasks (*Patel et al., 2022*). *Shankar et al. (2020)* synergic deep learning (SDL) model was developed to classify DR fundus images into different severity levels. This model was tested on the Messidor DR dataset. Future improvements could include incorporating filtering techniques to enhance image quality before processing. *Li et al. (2019)* proposed a novel deep neural network named OCTD-Net, specifically designed for early-stage DR classification using OCT images. The model was rigorously trained and evaluated on a comprehensive database of OCT images from Wenzhou Medical University (WMU), captured using a custom-built spectral domain OCT (SD-OCT) system. The significance of this work lies in its potential to assist ophthalmologists in evaluating and treating DR cases, thereby reducing the risk of vision loss. By enabling timely and accurate diagnosis, OCTD-Net demonstrates the promise of OCT images for cost-effective and time-efficient early-stage DR detection.

*Malhi, Grewal & Pannu (2023)* introduce an automated approach that precisely predicts the presence of exudates and microaneurysms in fundus images. These findings serve as essential indicators for grading DR, allowing us to determine whether it falls into the mild, moderate, or severe category. Future research plans to combine exudates and microaneurysms to enhance the grading process. Additionally, we aim to incorporate other relevant features, such as cotton wool spots and hemorrhages, for improved DR detection. Furthermore, expanding the dataset size will be a crucial step in refining this

approach. *Qureshi, Ma & Abbas (2021)* used the EyePACS dataset, sponsored by the California Healthcare Foundation, using 35,000 images. As a consequence, the utilization of diverse fundus images demonstrates that the innovative active deep learning (ADL) CNN architecture surpasses other methods in detecting lesions related to DR and accurately identifying the severity levels of DR. Furthermore, the ADL-CNN multi-layer architecture has the potential to extend its applications to various multimedia tasks, such as image dehazing, video tracking, and data mining. Several ML and DL algorithms are necessary. Improvements can be applied to the research findings by incorporating data augmentation and employing diverse preprocessing techniques to reduce noise and remove artifacts from the input photographs.

*Patil et al. (2023)* automate the detection of DR using DL through transfer learning. In this study, we achieved improved results by employing multiclass classification and enhancing generalization. Key techniques included appropriate pre-processing, data augmentation, and leveraging test-time augmentation (TTA). Despite dealing with an imbalanced dataset, our DL model demonstrated promising performance. *Monteiro (2023)* hybrid DL approach was developed by training separate DL models using a five-fold cross-validation technique and aggregating their predictions into a final score. To improve performance, researchers aim to improve the detection of lesions with mild and proliferative DR. Timely diagnosis of DR is essential, as it enables early treatment that can significantly reduce or avoid vision loss. Additionally, automatically detecting regions within the retinal image that may contain lesions could aid experts in their identification process. *Zago et al. (2020)* the CNNs were introduced, which has greatly impacted medical image analysis. The early diagnosis of DR using a CNN deep network technique in retinal images. *Das et al. (2021)* have utilized a small number of fundus images in their studies, often limited to a single dataset. *Sikder et al. (2021)* and *Xia et al. (2021)* concentrated on segmenting retinal lesions and evaluating the severity of DR based on the number of lesions in fundus images. Implementing medical imaging techniques requires computational resources, rigorous testing across diverse datasets, and real-time dataset collection from diabetic patients (*Nasajpour et al., 2022*; *Yu et al., 2021*). Maintaining a balance between data privacy and security is essential, particularly in compliance with the Health Insurance Portability and Accountability Act of 1996. As described in *Bonawitz et al. (2019)*, conscious system design decreases failures. This author used a deep network that captures local, global, and intermediate information, offering a more detailed and comprehensive understanding of fundus images from various clients (*Vishnu & Rajput, 2020*). Based on the background research, there are very few researchers focusing on DR classification using FL. This study presents a cutting-edge method for DR classification using FL with fundus images. Our proposal presents a federated learning (FL) framework that leverages a novel central server model to address the identified challenges. In FL, the central server is crucial in directing the training process and consolidating input from all participating clients. We preprocess and perform image augmentation on fundus images before server execution to enhance quality. Tests indicate that FL effectively generalizes fundus classification of images based on DR severity. Table 1 describes research on existing systems by considering parameters such as research approach, dataset, and challenges.

**Table 1 Research studies on existing systems.**

| Authors | Approach | Dataset | Challenges |
|---|---|---|---|
| Patil et al. (2023) | ResNet-50 DL model | EyePACS dataset and APTOS dataset | Restricted dataset and features |
| Abidin & Ismail (2022) | Hybrid model of SVM and KNN | Kaggle dataset | FL approaches also has smaller loss than the standard ML model |
| Nasajpour et al. (2022) | Standard transfer learning, FedAVG, and FedProx | Messidor, EyePACS, APTOS and IDRiD | Data access is prohibited under privacy regulations |
| Wang et al. (2023) | Federated uncertainty-aware aggregation paradigm (FedUAA) | Messidor, DDR, DRR, APTOS, and IDRiD | Improve collaborative DR staging performance by dynamically collecting reliable client data |
| Malhi, Grewal & Pannu (2023) | SVM and KNN | Messidor, DiaretDb and E-optha | Other features like cotton wool spots and hemorrhages will also be utilized in identifying DR. The dataset size will also improve |
| Sundar & Sumathy (2023) | Graph convolutional neural network (GCNN) | EyePACS dataset | In contrast, the graph model faces challenges with sharp edges and small image imperfections. |
| Ishtiaq, Abdullah & Ishtiaque (2023) | Fusing deep learning with local binary pattern (LBP) characteristics | EyePACS dataset | The method has the potential to detect retinal conditions like glaucoma, age-related macular degeneration (AMD), and cataracts. |
| Current approach | Examined five models (Resnet50, Densenet201, Alexnet, EfficientnetB7 and VGGnet19) | IDRiD | Lacks a discussion on explainable AI (XAI). |

## Deep learning

DL is a subset of artificial intelligence that is adept at detecting and classifying DR from fundus images. DL models can automatically extract relevant features, assign probabilities to different DR grades, and achieve high accuracy rates. This automation can aid in the early detection and prevention of vision loss associated with DR. The study introduces a DL method for automated DR classification using the microvascular structure of fundus images. It uses U-Net models for optic disc and blood vessel segmentation, followed by a hybrid CNN-SVD model for feature extraction and classification (*Sivapriya et al., 2024*). This approach identifies retinal biomarkers and achieves high accuracy on datasets like EyePACS-1, Messidor-2, and DIARETDB0, significantly improving DR detection and classification. The author presents a hybrid DL model for detecting DR by analyzing lesions in fundus images. It combines GoogleNet and ResNet models with an adaptive particle swarm optimizer (APSO) for feature extraction, followed by classification using ML models (*Jabbar et al., 2024*). The model achieves 94% accuracy on benchmark datasets, significantly improving precision, recall, accuracy, and F1 score for different DR severity levels. The study introduces a DL multistage training method for DR using stained retinal fundus images. It utilize models like InceptionResNetV2, VGG16, VGG19, DenseNet121, MobileNetV2, and EfficientNet2L. The training process involves extracting features with a customized classifier head, followed by fine-tuning. Data augmentation enhances model resilience and reduces overfitting (*Guefrachi, Echtioui & Hamam, 2024*). The method

InceptionResnetV2 achieves 96.61% accuracy on Kaggle's DR detection dataset. The article proposes an ensemble DL model to detect five severity levels of DR using fundus photographs. Initially, These CNNs are then combined to form an ensemble model, which is retrained on the dataset with five labels: No-DR, Mild, Moderate, Severe, and Proliferative-DR. The ensemble model achieves a validation accuracy of 87.31%, effectively classifying the severity of DR (*Kale & Sharma, 2023*).

The DL models used for detecting and classifying DR may have limitations, including restricted generalization, interpretability issues, concerns about preserving privacy, and sensitivity to data bias. To address these challenges, transfer learning techniques, strategies to improve model interpretability, and methods to mitigate data bias can be employed.

## Federated learning

Traditionally, the DL approach required consolidating data from multiple institutions into a single location for model training and testing. In contrast, the distributed learning paradigm known as FL allows participants to train models locally using their data. They then submit their updates to a central server, aggregating these modifications to improve the model's accuracy. This approach enhances the model's performance without requiring collaborators to directly access sensitive data or consolidate all collected data in one place (*Nguyen et al., 2022*; *Yarradoddi & Gadekallu, 2022*). Every institution stores its data locally; none is transferred or immediately accessed by another institution, as shown in Fig. 1. In the FL architecture, through the use of its training dataset, every institution trains a local model. After one training period, all local model parameters are then sent to the central server. The central server collects and aggregates all local parameters, which safely updates the global model. The updated model is then distributed back to each center for an additional training session. The global model is iterated until convergence occurs (*Supriya & Gadekallu, 2023*; *Thummisetti & Atluri, 2024*). The study highlights that FL models for DR classification are vulnerable to gradient inversion attacks, which can reconstruct sensitive patient data. The author used Bayesian Active Learning by Disagreement (BALD) score to identify images at risk. Results on the Fine-Grained Annotated Diabetic Retinopathy (FGADR) dataset show a negative correlation between the BALD score and image reconstruction quality, indicating that lower BALD scores mean higher susceptibility (*Nielsen, Tuladhar & Forkert, 2022*). The study emphasizes the privacy risks in FL for medical imaging. It proposes the BALD score to protect vulnerable data points, offering insights for privacy-preserving medical image analysis. The study uses FL to classify DR with OCT-angiography images, ensuring data privacy. Three institutions trained a VGG19 model locally and shared weights for central aggregation. The federated model performed comparably to individual models, maintaining classification accuracy while enhancing privacy and robustness (*Yu et al., 2021*). The study introduces a DL method for classifying retinal diseases using OCT images. It modifies three pre-trained models (DenseNet-201, InceptionV3, and ResNet-50) for feature extraction, optimized with ant colony optimization. The final classification uses k-nearest neighbors and support vector machines, achieving a high accuracy of 99.1%, outperforming existing techniques (*Subasi, Patnaik & Subasi, 2024*). The focus of these images was primarily on the region

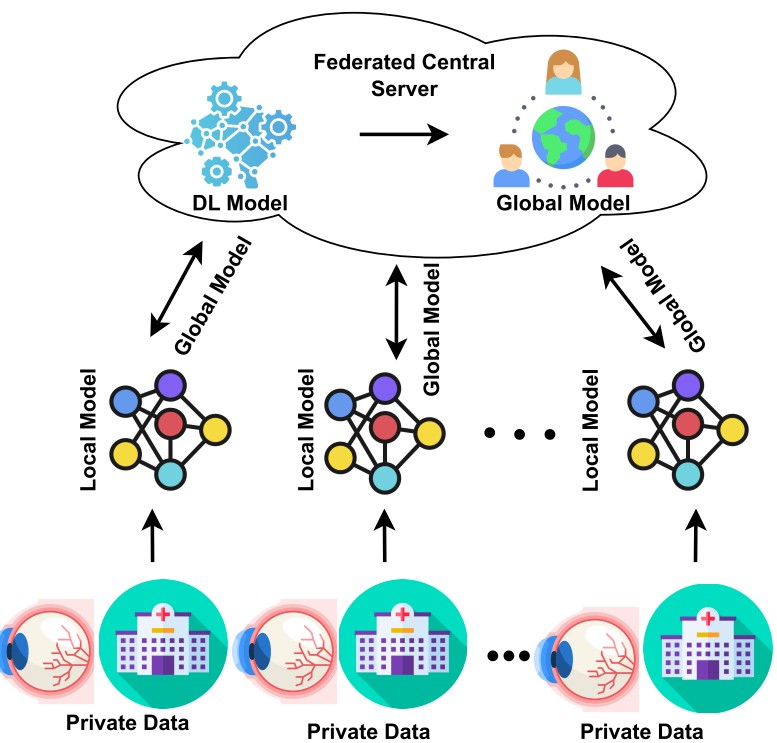

**Figure 1  FL concept in the healthcare sector.**     

near the macula. The study develops a privacy-preserving deep learning model for glaucoma detection using FL with OCT images. Seven eye centers participated, training models locally and sharing parameters for central aggregation. The FL model performed well across centers and on unseen datasets, ensuring patient privacy and data security (*Ran et al., 2024*).

In our study, we are introducing a new approach based on insights from related work. We propose a FedDL methodology for detecting and classifying DR using an FL environment. This methodology involves data aggregation, privacy preservation, model architecture selection, iterative model improvement, and inference and deployment. Our goal is to improve the diagnosis of DR while ensuring privacy and supporting collaborative healthcare efforts.

# PROPOSED METHODOLOGY

This section focuses on developing a FedDL model to identify DR and accurately assess its severity. Early detection is crucial for preventing vision loss associated with DR. We present the dataset description, the proposed FedDL method, federated averaging, and federated learning with model personalization.

## Dataset description

The proposed method examines the experimental results of DR identification using the IDRiD. A publicly accessible dataset created to study DR is called the IDRiD. A diabetes-related eye condition that damages the retina is called DR. The IDRiD dataset aims to aid

in the creation and assessment of algorithms for the detection and classification of DR (*Kalpana Devi & Mary Shanthi Rani, 2022*). The dataset contains retinal fundus images or pictures of the back of the eye. A significant challenge associated with publicly available healthcare datasets is their limited availability, often forcing researchers to work with the few accessible datasets. A publicly accessible dataset developed for DR research is called the IDRiD. A diabetes-related eye condition that damages the retina is called DR (*Porwal et al., 2018*). The dataset comprises retinal fundus images, capturing the back of the eye. Within this collection of 516 images, various pathological DR conditions are depicted. These images were captured using the Kowa VX-10 alpha digital fundus camera, which boasts a 50-degree field of view (FOV) (*Alyoubi, Shalash & Abulkhair, 2020*). The photos have a resolution of 4,288 × 2,848 pixels. To generalize the work, we used the STARE dataset, contains 400 fundus images captured using the Topcon TRV-50, which has a 35-degree field of vision (*Bali & Mansotra, 2024*).

## Dataset pre-processing

Raw data must be cleaned and arranged as part of the dataset preparation process before being utilized for training DL models. This procedure ensures that the data is in the right format and quality for efficient model training. These are a few typical stages in preparing datasets.

### Data augmentation

Data augmentation is a crucial technique utilized to enhance the diversity and quality of the dataset, particularly for tasks such as detecting and classifying DR using the IDRiD dataset. Below are some common data augmentation techniques applied to retinal images. In order to make the dataset more flexible, we are using augmentation techniques to balance it and prevent underfitting. These techniques involve horizontal and vertical flipping, rotation, zooming, and contrast enhancement. We rotate images by 360 degrees, flip them horizontally and vertically, zoom them to their original size, and enhance contrast to improve lighting. The augmented images are from train 890 and test 223. Data augmentation is an effective method in applications involving vision, such as the detection of DR, whereby the training dataset's diversity is artificially enhanced by applying different modifications to the pre-existing images. This enhances the generalization and resilience of the model. Given the scarcity of labeled images and the requirement that a model be invariant to specific transformations, data augmentation can be essential for detecting DR (*Naik et al., 2023*).

One of the main challenges in training DL models is the lack of sufficient and diverse datasets. DL models are characterized by numerous tunable parameters, which necessitate a proportional volume of data relative to the task's complexity. A mismatch in this proportionality can adversely impact the model's predictive accuracy and lead to overfitting, where the model fails to generalize to new data (*İncir & Bozkurt, 2024*). This issue is particularly pronounced in domains such as plant or human disease classification, where data collection is inherently challenging. Moreover, even with adequate data, the

class imbalance can pose a significant problem, as some classes often have abundant data while others do not (*Liu et al., 2024*). Data augmentation techniques are employed to synthetically enhance the dataset to combat these issues. This process involves applying various transformations—such as cropping, rotation, flipping, translation, scaling, adjusting color spaces, injecting noise, varying brightness, and altering color balance—to generate new data from existing samples. The goal is to ensure a diverse and ample dataset for robust model training (*Prabhakar et al., 2024*).

## Federated averaging

Federated averaging is a powerful technique used in FL to detect and classify DR, a consequence of diabetes that causes lesions on the retina. This condition can potentially lead to vision impairment or blindness. Detecting DR early is crucial to prevent irreversible vision loss. The primary goal of our work is to develop a model capable of detecting and classifying DR to prevent its progression to a severe stage. We propose using the FedDL technique, which involves sending the best score (parameters or weights) to the server. Before we talk about our proposed methodology, we will look over the standard FedAvg algorithm. FL follows the sequence outlined in the Algorithm 1. In line 4, the client selected for the current round is chosen. Lines 5 and 6 describe the process of receiving weights from various clients. Line 8 calculates the weighted average to determine the global weights. After aggregation, the global model is sent from the server to the client, and this cycle repeats until an optimal model is achieved.

In FL, models are trained locally, allowing the model to access client data without transferring it to a central server. However, data sensitivity often makes compiling broad and varied datasets challenging, which is crucial for building robust DL models. FL addresses this challenge by decentralizing the training of ML models (*Chetoui & Akhloufi, 2023*). The approach used for detecting DR utilizes the weighted average of models, commonly known as Fedavg. Initially, a global model is established. In each round t, the central server sends the current global model wt to a selected subset C of all participating institutions K. These selected institutions are denoted by the set St. After training the model on its local data Pk, Each institution, denoted as 'k', updates its local model parameters and forwards them to a central server, which then integrates these contributions to construct a model wt + 1k (*Riedel et al., 2023*). The server then creates a new global mode by combining the weights of the incoming models using the Eq. (1).

$$w_{t+1} = \sum_{k=1}^{n} e^{S_t} \cdot w_k \cdot n_k \cdot n_t. \tag{1}$$

In this, nt denotes the total number of samples across all institutions, while nk represents the number of samples at each institution k. Once the training process concludes, the server distributes the aggregated model to all institutions connected to the network. The FL framework is an iterative approach that partitions the development of a global central DL model while preserving client privacy. It operates in rounds, where each

**Algorithm 1  Federated average algorithm (*Supriya & Gadekallu, 2023*).**

1:  **function** SERVERAGGREGATION($\eta N$)
2:      Initialise $w_0$
3:      **for** every iteration $a = 1, 2, \ldots$ **do**
4:          $S_a \leftarrow$ (clients are chosen randomly from set of $\max(C_K, 1)$)
5:          **for** every client $t \in S_a$ in parallel **do**
6:              $w_{k,a+1} \leftarrow$ UPDATECLIENT($k, w_a$)
7:          **end for**
8:          $w_{a+1} \leftarrow$ average of the weights that are collected $w_{k,a+1}$ of $S_a$ clients
9:      **end for**
10: **end function**
11: **function** UPDATECLIENT($k, w$)
12:     Continue the learning process on client $t$ with weight $w$ until the client completes the task
13:     the client arrives at $E$ epochs
14:     Update the weight accordingly after the learning phase
15:     **return** $w$ to the server
16: **end function**

round involves interactions between clients and the server. The fundamental premise is that a predefined set of clients exists, each with local private data dk kept separate from other clients and the server. That is kept separate from the server and other clients. The number of rounds be represented by t = 1, 2,…, T. At the start of each round, a group of clients m is selected, and the server communicates the current state of the global algorithm to these clients (*Matta et al., 2023*). A prominent FedAvg method trains a global model across several decentralized devices or servers without transferring raw data in FL. Instead of examining the proposed approach, we will examine the FedAvg algorithm as it is standard (*Sun, Li & Wang, 2022*). The FedAvg algorithm's basic steps are in Algorithm 1.

## The proposed FedDL method

FL is a decentralized approach where multiple clients, such as devices or institutions, collaborate to train a model without sharing their local data. Instead, they share model updates, like gradients or weights, with a central server that aggregates these updates to enhance a global model. The clients, which can be devices or institutions with local datasets and computational power, train the models, while the server is a central entity that coordinates the training process and aggregates model updates. Various DL models, such as ResNet50, DenseNet201, AlexNet, EfficientNetB7, and VGG19, are used. These models are pre-defined and chosen based on the task requirements and computational constraints. A brief explanation of the proposed framework's working process follows.

**Input and output:**

**Input** The IDRiD dataset, a retinal image dataset annotated for DR classification tasks.

**Output** The best-performing model for DR classification, based on key performance metrics such as accuracy, AUC, precision, and recall.

**Loading and preprocessing the IDRiD Dataset:**

Image preprocessing includes resizing images to match the input requirements of DL models and normalizing pixel values to a range of 0 to 1 for improved model convergence. Data augmentation techniques such as random rotations, flips, zooms, and brightness adjustments are applied to the training data to enhance model robustness and prevent overfitting. The dataset is divided into training and test sets, with the training set distributed across clients in a FL environment.

**Deep learning model initialization:**

Several pre-existing deep learning architectures are initialized, including ResNet50, DenseNet201, AlexNet, EfficientNetB7, and VGG19, each chosen for its ability to handle image classification tasks effectively. ResNet50 is known for its deep architecture and use of skip connections, DenseNet201 promotes feature reuse with densely connected layers to improve accuracy, AlexNet is a simpler, early CNN architecture with proven success in image classification, EfficientNetB7 is known for compound scaling that balances performance and efficiency, and VGG19 is a deep network with a simple architecture effective in image-based tasks.

**Federated learning setup:**

In the FL environment, the dataset is divided among $N$ clients, with each client receiving a distinct subset of the training data to ensure that the data is not stored in a central location. The global model is initialized on the server to match the architecture of the client models, and its weights are updated step by step during the training process.

**FedDL process:**

The training process in federated learning consists of multiple rounds (1 to $R$). In each round, clients receive the current global model from the server and train it locally on their data using optimization techniques such as stochastic gradient descent (SGD). This local training allows the model to learn from the distributed data without needing to access data from other clients. Once training is complete, clients send their updated model weights (gradients or parameters) back to the server. The server then combines these weights using FedAvg, which calculates a weighted average of the models based on the size of each client's dataset (Eq. (2)).

$$\mathbf{w} = \sum_{i=1}^{K} \frac{n_i}{N} \mathbf{w}_i \tag{2}$$

where: $\mathbf{w}_i$ are the model weights from client $i$, $n_i$ is the number of samples in client $i$'s dataset, $K$ is the number of participating clients, and $N$ is the total number of samples across all clients according to Eq. (3).

$$N = \sum_{i=1}^{K} n_i \qquad (3)$$

The global model is updated with the aggregated weights, enabling the server to utilize the combined insights from the distributed datasets.

**Model evaluation:**

After several rounds of federated training, the global model is evaluated on a centralized test set, unseen during the training process. Performance is assessed using the following metrics. Accuracy refers to the proportion of correctly classified diabetic retinopathy (DR) cases, while precision indicates the percentage of correctly predicted positive DR cases. Recall measures the model's ability to detect true positive DR cases, and the F1 score, which is the harmonic mean of precision and recall, provides a balanced view of the model's performance. Sensitivity, similar to recall, reflects the model's ability to identify true positive DR cases, whereas specificity measures the model's ability to correctly identify true negative DR cases. Lastly, the Area Under the Curve (AUC) reflects the model's ability to distinguish between DR-positive and DR-negative classes.

**Model comparison and selection:**

The global models based on ResNet50, DenseNet201, AlexNet, EfficientNetB7, and VGG19 are compared based on the performance metrics mentioned above. The model with the highest AUC or accuracy is selected as the best-performing model for DR classification.

The proposed FedDL framework allows for effective diabetic retinopathy detection and classification across distributed institutions, enhancing privacy and security by keeping data local. Using powerful deep learning models in combination with federated learning, this framework ensures high accuracy while addressing the limitations of centralized model training. This approach is particularly valuable in healthcare scenarios, such as DR detection and classification, where privacy concerns and regulations limit the sharing of sensitive medical data. The proposed model states the FL in DR detection and classification in Fig. 2.

The proposed framework Algorithm 2 outlines a comprehensive approach for classifying DR using DL models in an FL environment. The algorithm is divided into several key steps, beginning with data pre-processing and augmentation using the IDRiD dataset. It then involves initializing pre-trained DL models such as ResNet50, DenseNet201, AlexNet, EfficientNetB7, and VGG19, and setting hyperparameters for each model. The algorithm also describes the setup for FL in a multi-client environment, including the initialization of a global model and the FL process using FedAvg. After FL, the final global model is evaluated on the test dataset, and a comparison of the performance of the different DL models is conducted to select the best-performing model based on evaluation metrics.

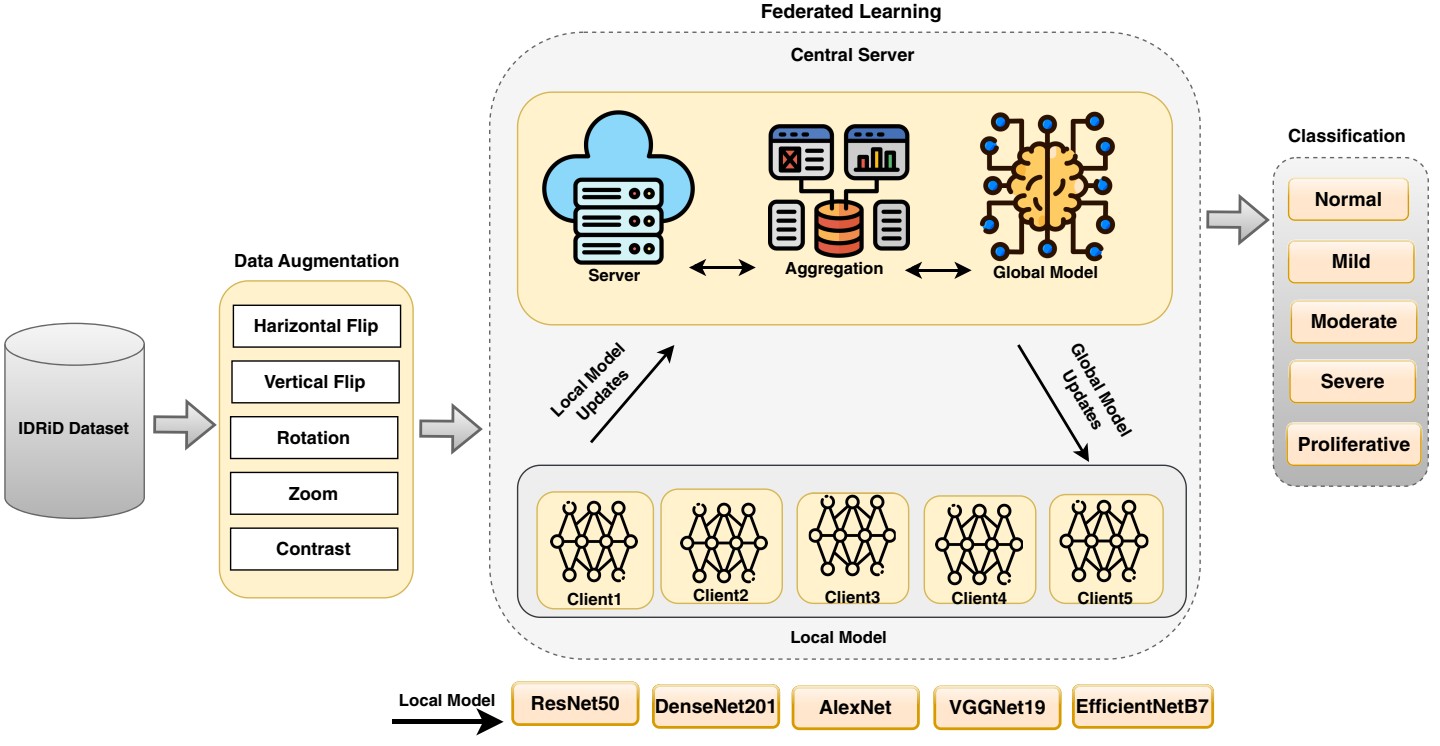

**Figure 2** A proposed FedDL architecture for DR detection and classification.

## Federated learning with model personalization

FL is a decentralized ML approach in which multiple clients collaborate to train a global model while preserving the privacy of their raw data. DL models can be effectively utilized within the FL environment. Each client trains DL models locally on their datasets, and the global model is updated by aggregating the individualized model updates from all clients.

**ResNet50:** Using for DR detection and classification involves leveraging. The ResNet-50 architecture is used for the specific task of identifying signs of DR in retinal images. DR is a medical condition associated with diabetes that affects the retina, and early detection is crucial for timely intervention. ResNet-50 is a deep CNN architecture that includes 50 layers. It is characterized by using residual blocks, which contain shortcut connections to mitigate the vanishing gradient problem (*Karthika, Durgadevi & Rani, 2023*). ResNet50 has shown effectiveness in training very deep networks. ResNet50 is often pre-trained on large-scale image datasets, such as ImageNet. This pre-training allows the model to learn general features from diverse images before being fine-tuned for the specific task of DR detection.

**Densenet201:** Similar to other DenseNet architectures, DenseNet201 is a deep neural network suitable for detecting and classifying DR. DenseNet, which stands for densely connected convolutional networks, is characterized by densely connected blocks where each layer receives input from all preceding layers. These dense connections promote feature reuse, enhance gradient flow, and contribute to more efficient training. Below is a

| Algorithm 2 | Proposed framework algorithm for DR classification using FL. |
|---|---|

**Input:** IDRiD Dataset

**Output:** Best-performing model for DR classification

1: **Load and preprocess the IDRiD dataset**

2: Resize images to fit model input size

3: Normalize pixel values between 0 and 1

4: Split the dataset into training, and test sets

5: **Apply data augmentation to training data**

6: Augment images with random rotations, flips, zoom, and brightness adjustments

7: **Initialize DL models: ResNet50, DenseNet201, AlexNet, EfficientNetB7, VGG19**

8: **Distribute the dataset to N clients in the FL environment**

9: Each client receives a subset of the training data

10: **Initialize the global model with the same architecture as client models**

11: **for** each round of FL (1 to R) **do**

12:     **for** each client (1 to N) **do**

13:         Receive the global model from the server

14:         Train the model locally on the client's dataset

15:         Send the updated model weights to the server

16:     **end for**

17:     **Server-side aggregation:**

18:     Aggregate the model weights from all clients using Federated Averaging

19:     Update the global model with the aggregated weights

20: **end for**

21: **Evaluate the global model on the centralized test dataset**

22: Calculate performance metrics: Accuracy, Precision, Recall, F1 score, specificity, sensitivity, AUC-ROC

23: **Compare the performance of the models (ResNet50, DenseNet201, AlexNet, EfficientNetB7, VGG19)**

24: Select the best model based on performance metrics (highest AUC or accuracy)

concise overview of how DenseNet201 can be utilized for DR detection (*Dinpajhouh & Seyyedsalehi, 2023*). DenseNet-201 is a specific variant of the DenseNet architecture with 201 layers. It consists of densely connected blocks that contain bottleneck layers and skip connections, fostering feature reuse and mitigating the vanishing gradient problem.

**AlexNet:** is a pioneering CNN that rose to prominence due to its victory in the ImageNet Large Scale Visual Recognition Challenge (ILSVRC) in 2012. This achievement marked a significant breakthrough in the field of DL. Even though it was one of the first DL architectures, more sophisticated models like VGG, ResNet, and others have supplanted it. Nevertheless, AlexNet can still be used for DR detection and classification, particularly in cases where computational resources are limited. Here is a brief overview: AlexNet is structured with five layers of convolution followed by a sequence of three layers that are

fully connected (*Das, Biswas & Bandyopadhyay, 2023*). It introduced the use of ReLU activation functions and dropout for regularization. Max-pooling is applied after the first and second convolutional layers.

**EfficientB7:** is a variant of the EfficientNet architecture introduced to achieve better model performance while maintaining computational efficiency. Efficient-NetB7 is one of the larger models in the EfficientNet family, and its use in DR detection and classification involves leveraging its scalability and generalization capabilities (*Giroti et al., 2023*). Here is a brief overview: EfficientNetB7, like other EfficientNet models, employs a compound scaling approach to ensure a harmonious balance between its depth, width, and resolution. The architecture includes multiple blocks of convolutional layers, such as MBConv (Mobile Inverted Residual Bottleneck) blocks, designed for efficiency.

**VGGNet19:** Visual geometry group network (VGG) is a CNN architecture that gained attention for its simplicity and effectiveness. VGGNet19, a variant of VGGNet, comprises 19 layers, including 16 convolutional layers and three fully connected layers (*Rakesh et al., 2023*). While VGGNet is an older architecture compared to more recent models like ResNet and EfficientNet, it can still be applied to DR detection and classification. Here's a brief overview. VGGNet19 comprises 16 convolutional layers, each accompanied by a rectified linear unit (ReLU) activation function and three fully connected layers. The convolutional layers have small $3 \times 3$ filters, and max-pooling is applied after some convolutional blocks.

We have evaluated five different models for use in an FL setting to identify and categorize DR based on their strengths in processing complex image data. The deep architecture and residual connections of ResNet50 can help address the vanishing gradient issue, while DenseNet201's dense connections facilitate maximum information flow between layers, making it ideal for extracting features from medical images. AlexNet's straightforward yet effective approach to image classification tasks also makes it a strong candidate, and EfficientNetB7's balanced model size and accuracy through scaling depth, width, and resolution are advantageous. Furthermore, VGGNet19's simplicity and depth make it suitable for capturing detailed features in images, which is essential for DR classification in an FL environment.

## RESULTS AND DISCUSSION

In the following section, we present the results obtained from our proposed model, comparing them with conventional FL methods that rely on centralized training. We outline the experimental setup, dataset collection and preparation, hyperparameters and tuning strategies, performance metrics, experimental results, performance of the proposed approach, analysis and discussion, evaluation of performance using a different dataset, and strengths and weaknesses in real-world clinical settings.

### Experimental setup

The implementation was done in Python on a Windows 11 Pro PC with an Intel i9 core processor. The system runs on a 64-bit operating system with 32.0 GB of RAM and an NVIDIA GeForce MX150 GPU. The experiments were conducted over 100 iterations

spanning 10 epochs. We utilized Keras version 2.4.3 and TensorFlow version 2.3.0 to implement FL. Our approach focuses on improving communication efficiency in FL scenarios. Specifically, we utilize the FedAvg algorithm and five DL models performed in the FL environment. We iteratively update weights transmitted from clients to the server, improving overall performance.

## Dataset acquisition and preparation

This work utilized an image dataset sourced from Kaggle (https://www.kaggle.com/datasets/aaryapatel98/indian-diabetic-retinopathy-image-dataset), accessed on 11 July 2023. Specifically, We accessed the IDRiD, which comprises a diverse collection of 516 images. These images were divided into training and testing sets with an 80:20 ratio after augmentation, specifically for DR detection and classification. IDRiD stands out as the first database representing an Indian population, providing unique pixel-level annotations for both typical DR lesions and normal retinal structures. Furthermore, each image includes detailed information on the severity of DR.

## Hyperparameters and tuning strategies

Hyperparameters are settings or parameters that are determined before the training of a machine learning algorithm and are not learned from the data during training. These settings control the behavior of the algorithm. Common hyperparameters in machine learning include the learning rate, which determines how quickly the model updates its parameters during training; the number of epochs, which is the number of times the entire dataset is passed through the model during training; batch size, which is the number of samples processed at once during training; regularization strength, which controls the amount of regularization applied to prevent overfitting; and network architecture, which includes the number of layers, neurons per layer, and activation functions in the neural network.

**Random search.** It involves randomly sampling hyperparameter values from a specified distribution and evaluating the model's performance for each set. This process is repeated multiple times, and the best hyperparameters are selected based on the evaluation results.

## Performance metrics

The performance of the proposed model is evaluated using metrics such as accuracy, precision, recall, specificity, F1 score, sensitivity, receiver operating characteristic (ROC), and area under the curve (AUC). These metrics are defined in Eqs. (4) through (12).

**Accuracy:** Accuracy quantifies the correctness of predictions for a given dataset, with values ranging from 0 to 1.

$$\text{Accuracy} = \frac{TP + TN}{TP + TN + FP + FN} \tag{4}$$

**Precision:** Precision gauges the accuracy of predictions made by the classifier, with values ranging from 0 to 1.

$$\text{Precision} = \frac{TP}{TP + FP} \tag{5}$$

**Recall:** Recall assesses the classifier's ability to capture relevant predictions, with values ranging from 0 to 1.

$$\text{Recall} = \frac{TP}{TP + FN} \tag{6}$$

**Specificity:** Specificity refers to the capacity to accurately recognize true negatives in a classification system. It is quantified on a scale from 0 to 1.

$$\text{Specificity} = \frac{TN}{TN + FP} \tag{7}$$

**F1 score:** The F1 score leverages recall to determine the proportion of true positive records out of the total actual positive records.

$$\text{F1 score} = \frac{2 \cdot \text{Precision} \cdot \text{Recall}}{\text{Precision} + \text{Recall}} * 100 \tag{8}$$

**Sensitivity:** Sensitivity, also referred to as the true positive rate or recall, is a crucial metric in ML. It quantifies the proportion of actual positive cases that a model correctly identifies.

$$\text{Sensitivity} = \frac{TP}{TP + FN} \tag{9}$$

**ROC:** The ROC curve graphically represents the trade-off between the true positive rate (TPR) and the false positive rate (FPR) across different threshold settings. True positive rate (TPR) is defined as:

$$\text{TPR} = \frac{TP}{TP + FN} \tag{10}$$

False positive rate (FPR) is defined as:

$$\text{FPR} = \frac{FP}{FP + TN} \tag{11}$$

**AUC:** The AUC, which stands for the area under the ROC curve, provides a single metric that summarizes the model's overall performance. An AUC value of 1 indicates a perfect model, while 0.5 suggests a random classifier. The AUC is calculated as the integral of the ROC curve:

$$\text{AUC} = \int_0^1 \text{TPR}(t) d(\text{FPR}(t)) \tag{12}$$

where TP (true positives), TN (true negatives), FP (false positives), and FN (false negatives)

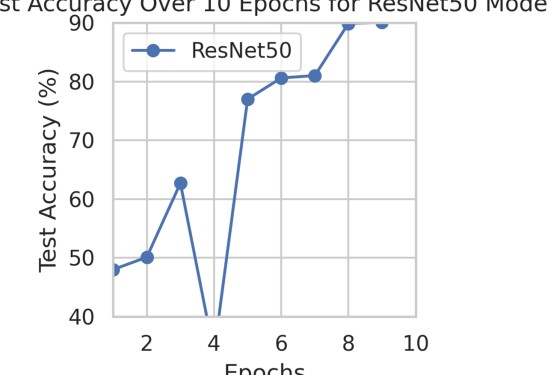

**Figure 3 Test accuracy for the proposed model.**

are defined, the classifier with the highest precision, recall, F-measure, and accuracy is considered the best predictive classifier.

## Experimental results

This section outlines the experiments performed to evaluate and analyze DL models in a FL setting. Subsequently, we examine the performance of the FedAvg-enabled DL approach on the IDRiD dataset and STARE and compared it with conventional DL models.

### *Performance of the proposed approach*

This component evaluates the training and performance evaluation of the model. The samples from the IDRiD dataset are initially used to train the server model. The server model is then assigned to the clients. Typically, we evaluate the model's performance with five clients. We decided on 0.0025 as the learning rate number. Random selection is used to select observations for every client device in the collection. The accuracy results are depicted in Fig. 3, showing that the proposed model's correctness improves over time. The model's accuracy starts low but steadily increases as the proposed approach learns, making it more reliable in detecting and classifying DR images. Figure 4 illustrates the loss graph of the proposed model, indicating a continuous decrease in loss over time. Although the model's loss graph initially appears large, it gradually diminishes as the proposed approach learns, enhancing its stability and competency in detecting and classifying DR images. The test accuracy for the proposed model is shown in Fig. 3 over the 10 epochs for the resnet50 model, and the loss is depicted for the proposed model in Fig. 4. Additionally, Fig. 5 illustrates the test accuracy for the other four different models, while Fig. 6 displays the loss for those models. The proposed work is compared with existing research regarding accuracy, dataset, and technology. Author-wise details are provided in the following Table 2. Table 3 illustrates the performance of the DenseNet model within a FL framework. Table 4 describes the performance of the AlexNet model in the FL environment. Table 5 describes the performance of the EfficientNetB7 model in the FL environment. Table 6 describes the performance of the VGGNet19 model in the FL environment.

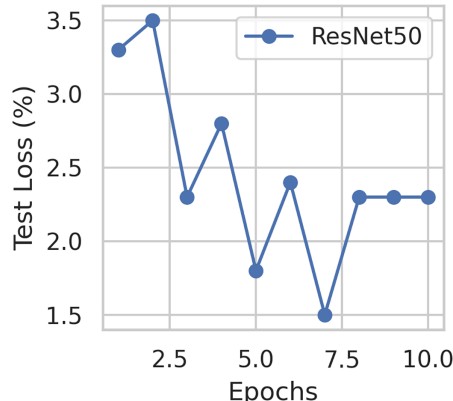

**Figure 4 Test loss for the proposed model.**

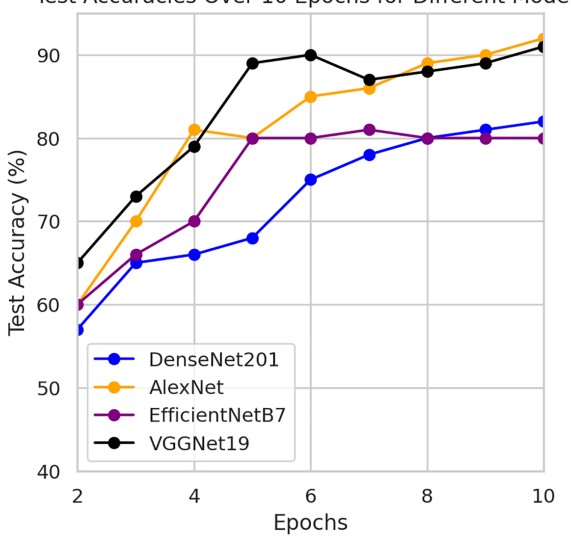

**Figure 5 Test accuracy for other four models.**

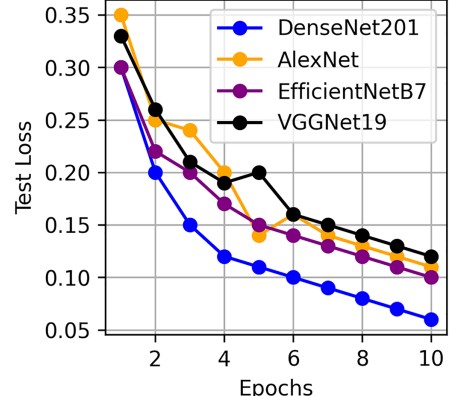

**Figure 6 Test loss for other four models.**

**Table 2 Comparing the performance of the proposed work with existing DR detection methods.**

| Work | Acc. % | Dataset | Technique used |
|---|---|---|---|
| Proposed work | 94.66% | IDRiD | FedDL |
| *Sunkari et al. (2024)* | 93.51% | APTOS | Resnet18 |
| *Saranya & Prabakaran (2020)* | 90.29% | IDRiD | CNN |
| *Nasajpour et al. (2022)* | 86.52% | IDRiD | FL approach |
| *Goswami, Ashwini & Dash (2023)* | 82.13% | IDRiD | InceptionV3 model |
| *Chetoui & Akhloufi (2023)* | 71.00% | IDRiD | Vision transformer architecture |

**Table 3 Performance metrics for the DenseNet201 model in a FL environment.**

| Performance metrics | Pre. % | Rec. % | F1. % | Spec. % | Sen. % |
|---|---|---|---|---|---|
| Epoch1 | 95% | 95% | 95% | 95% | 95% |
| Epoch2 | 94% | 94% | 94% | 94% | 94% |
| Epoch3 | 91% | 91% | 91% | 91% | 91% |
| Epoch4 | 90% | 90% | 90% | 90% | 90% |
| Epoch5 | 88% | 88% | 88% | 88% | 88% |
| Epoch6 | 87% | 87% | 87% | 87% | 87% |
| Epoch7 | 86% | 86% | 86% | 86% | 86% |
| Epoch8 | 85% | 85% | 85% | 85% | 85% |
| Epoch9 | 86% | 86% | 86% | 86% | 86% |
| Epoch10 | 90% | 90% | 90% | 90% | 90% |

**Table 4 Performance metrics for the AlexNet model in a FL environment.**

| Performance metrics | Pre. % | Rec. % | F1. % | Spec. % | Sen. % |
|---|---|---|---|---|---|
| Epoch1 | 87% | 87% | 87% | 87% | 87% |
| Epoch2 | 92% | 92% | 92% | 92% | 92% |
| Epoch3 | 88% | 88% | 88% | 88% | 88% |
| Epoch4 | 87% | 87% | 87% | 87% | 87% |
| Epoch5 | 92% | 92% | 92% | 92% | 92% |
| Epoch6 | 92% | 92% | 92% | 92% | 92% |
| Epoch7 | 89% | 89% | 89% | 89% | 89% |
| Epoch8 | 88% | 88% | 88% | 88% | 88% |
| Epoch9 | 88% | 88% | 88% | 88% | 88% |
| Epoch10 | 88% | 88% | 88% | 88% | 88% |

## Performance comparison with other techniques

To evaluate the effectiveness of our proposed FL framework, we conducted experiments using various nature-inspired optimization techniques, including Ant Colony Optimization (ACO) and Artificial Bee Colony (ABC). These experiments were conducted

**Table 5 Performance metrics for the EfficientNetB7 model in a FL environment.**

| Performance metrics | Pre. % | Rec. % | F1. % | Spec. % | Sen. % |
|---|---|---|---|---|---|
| Epoch1 | 90% | 90% | 90% | 90% | 90% |
| Epoch2 | 92% | 92% | 92% | 92% | 92% |
| Epoch3 | 91% | 91% | 91% | 91% | 91% |
| Epoch4 | 92% | 92% | 92% | 92% | 92% |
| Epoch5 | 85% | 85% | 85% | 85% | 85% |
| Epoch6 | 89% | 89% | 89% | 89% | 89% |
| Epoch7 | 90% | 90% | 90% | 90% | 90% |
| Epoch8 | 89% | 89% | 89% | 89% | 89% |
| Epoch9 | 90% | 90% | 90% | 90% | 90% |
| Epoch10 | 90% | 90% | 90% | 90% | 90% |

**Table 6 Performance metrics for the VGGNet19 model in a FL environment.**

| Performance metrics | Pre. % | Rec. % | F1. % | Spec. % | Sen. % |
|---|---|---|---|---|---|
| Epoch1 | 88% | 88% | 88% | 88% | 88% |
| Epoch2 | 88% | 88% | 88% | 88% | 88% |
| Epoch3 | 88% | 88% | 88% | 88% | 88% |
| Epoch4 | 88% | 88% | 88% | 88% | 88% |
| Epoch5 | 88% | 88% | 88% | 88% | 88% |
| Epoch6 | 88% | 88% | 88% | 88% | 88% |
| Epoch7 | 89% | 89% | 89% | 89% | 89% |
| Epoch8 | 89% | 89% | 89% | 89% | 89% |
| Epoch9 | 89% | 89% | 89% | 89% | 89% |
| Epoch10 | 87% | 87% | 87% | 87% | 87% |

on the IDRiD dataset. Figure 7 provides a comprehensive comparison of the performance metrics achieved by our proposed methodology with the ACO and ABC techniques. The ROC curve is a graph that shows the performance of a classification model by plotting the false positive rates against the true positive rates. It is effective for detecting the presence or absence of a disease. For accurate detection, a low false positive rate and a high true positive rate are essential. According to the Fig. 8, the proposed method achieves a true positive rate close to 0.93 at lower false positive rates. This indicates that the proposed method classifies the disease with higher accuracy. In comparison, the other techniques, ACO and ABO, showed AUC values of 0.88 and 0.87, respectively.

## Analysis and discussion

Our proposed framework achieved impressive results in detecting and classifying DR, outperforming traditional centralized approaches in terms of accuracy and AUC. By leveraging data augmentation and FedAvg, we enhanced model generalization and reduced overfitting, while nature-inspired algorithms for hyperparameter optimization

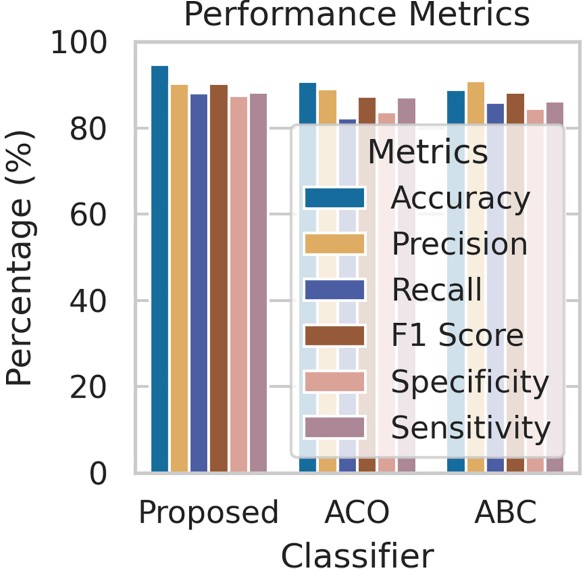

**Figure 7** The performance comparison with other techniques.

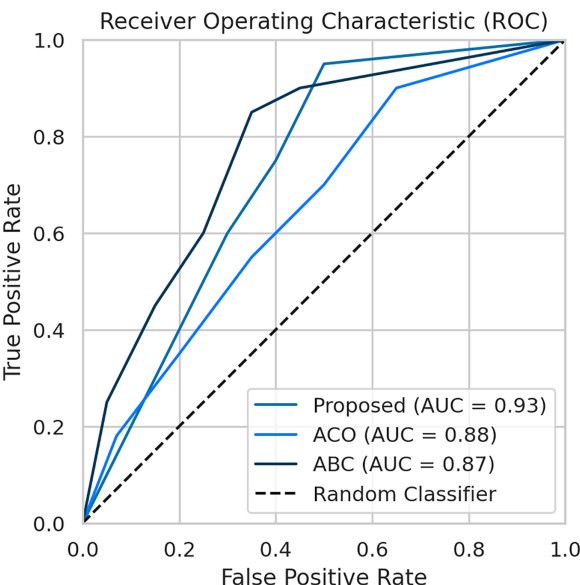

**Figure 8** The AUC-ROC performance comparison with other techniques.

further boosted performance. This approach allowed us to train models on distributed datasets without compromising patient privacy, a significant advantage in healthcare settings where data sharing is challenging. In this work, we compare our proposed approach to existing methods regarding the dataset and methodology employed for detecting and classifying diabetic retinopathy. Additionally, we critically examine the limitations identified in the literature. Table 2 provides a comprehensive comparison with the current method, demonstrating that our proposed approach yields significant

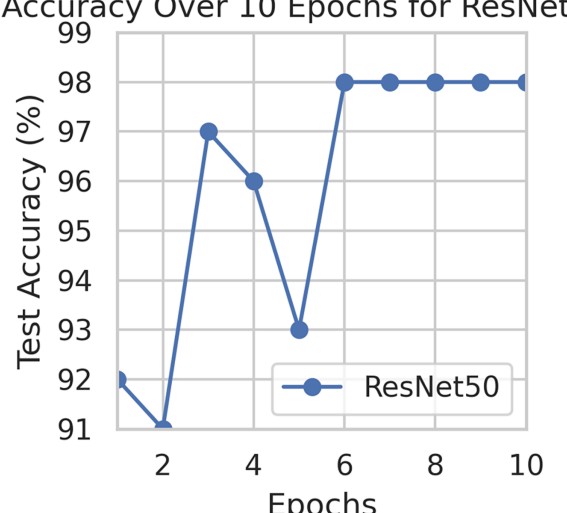

Test Accuracy Over 10 Epochs for ResNet50 Model

**Figure 9 Test accuracy for the ResNet50 model on the STARE dataset.**

improvements. Furthermore, we present the simulation results obtained from our proposed diabetic retinopathy detection model, evaluated on a dataset comprising 1,113 fundus images after augmentation with train and test sets. To gauge the model's efficiency, we assess a set of performance measures, including accuracy, sensitivity, specificity, recall, precision, F1 score, and AUC-ROC. Despite these promising results, future research could explore different FL algorithms and investigate the impact of varying data distributions across clients to gain further insights into the effectiveness of this approach.

## Evaluate the performance using a different dataset

To generalize the robustness of our proposed methodology using FedAvg. We have conducted additional experiments using the STARE dataset. In our study, we focused on detecting DR using the STARE dataset. The model exhibited a remarkable upward accuracy trend throughout multiple training rounds. Initially, in the first round, the accuracy stood at 92%, but as we progressed, it steadily improved. By the sixth round, the model achieved a peak accuracy of 98%. Simultaneously, the corresponding loss consistently declined, indicating successful model optimization over consecutive rounds. This iterative process allowed us to fine-tune the model and enhance its performance in identifying early signs of DR from retinal images. Figures 9 and 10 illustrates the accuracy and loss performance of the ResNet50 model using the STARE dataset.

## Strengths and weaknesses in real-world clinical settings

To facilitate a more thorough discussion, it is essential to address both the strengths and weaknesses of our work on the detection and classification of DR using DL models in a FL environment.

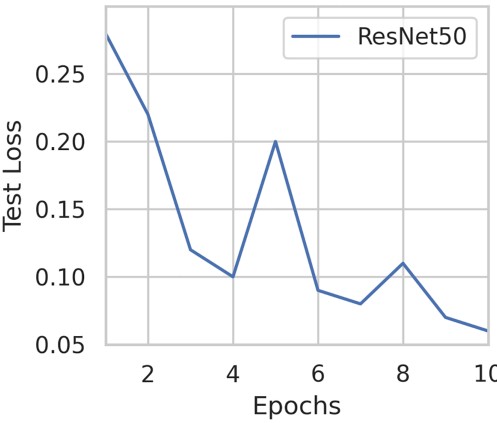

Test Loss Over 10 Epochs for ResNet50 Model

**Figure 10 Test loss for the ResNet50 model on the STARE dataset.**

## Strengths

**Data privacy:** FL enhances patient data privacy by keeping data localized and only sharing model updates.

**Generalization:** The model benefits from diverse data sources, improving its ability to generalize across different populations.

**Scalability:** FL allows for scalable model training across multiple institutions without the need for centralized data storage.

**Performance:** The integration of advanced techniques like transfer learning and privacy-preserving methods can maintain high model accuracy.

## Weaknesses

**Data heterogeneity:** Variability in data quality and distribution across institutions can affect model performance and consistency.

**Communication overhead:** FL requires significant communication between nodes, which can be resource-intensive and slow down training.

**Complexity:** Implementing FL involves complex coordination and management of multiple models and data sources.

**Ethical and legal challenges:** Ensuring compliance with ethical guidelines and legal frameworks for patient data privacy can be challenging and may vary across regions.

## CONCLUSION AND FUTURE DIRECTIONS

The proposed methodology FedDL implements the ResNet50, Densenet201, AlexNet, EfficientNetB7, and VGGNet19 models in a FL environment. In this environment, multiple clients train their local models on their data and update them to the central server. The central server aggregates the client data and sends it to the global model. The IDRiD dataset is used for image augmentation during the training of the client models. Simple DL models are used as the base model for each client, and their performance on a validation set is evaluated. The accuracy of each client's model on the test set is then used to update their

local models. The primary objective of this system is to benefit multiple medical facilities. To address privacy concerns, FL was adopted—an approach that shares model parameters without exposing actual patient data. Using the FedDL framework, the study investigated five key DL models capable of distinguishing DR. The FL approach's overall performance is assessed based on the highest accuracy achieved by the Resnet50 model out of the five models on the test set.

In the field of medical imaging, FL raises ethical concerns regarding patient data privacy. To address this, it is important to minimize shared data, implement secure aggregation, enforce strict access controls, and obtain patient consent. These measures are essential to ensure responsible and ethical FL, thereby protecting patient privacy and promoting trust. Specific measures include using techniques like FedAvg, training models locally, encrypting model updates, conducting regular security assessments, and communicating the process and its benefits to patients.

## Future Directions

Future studies on detecting and classifying DR using DL models in an FL environment could explore several research questions and hypotheses. These include examining how data heterogeneity across institutions affects model performance, identifying effective model aggregation techniques, and integrating privacy-preserving methods like differential privacy without compromising accuracy. Additionally, investigating the role of transfer learning in enhancing model performance across different populations, developing real-time adaptation mechanisms for incorporating new data, and addressing ethical and legal implications are crucial. These directions provide a detailed roadmap for advancing federated learning in DR detection.

### Funding
The authors received no funding for this work.

### Competing Interests
The authors declare that they have no competing interests.

### Author Contributions
- Dasari Bhulakshmi conceived and designed the experiments, performed the experiments, analyzed the data, performed the computation work, prepared figures and/or tables, authored or reviewed drafts of the article, and approved the final draft.
- Dharmendra Singh Rajput conceived and designed the experiments, performed the experiments, analyzed the data, performed the computation work, prepared figures and/or tables, authored or reviewed drafts of the article, and approved the final draft.

### Data Availability
The code is available at GitHub and Zenodo:

- https://github.com/dasari2023/Code

- Dasari, B. (2024). FedDL: Personalized Federated Deep Learning for Enhanced Detection and Classification of Diabetic Retinopathy [Data set]. Zenodo. https://doi.org/10.5281/zenodo.14178786.

The third party dataset used is available at Kaggle (Aarya Patel): https://www.kaggle.com/datasets/aaryapatel98/indian-diabetic-retinopathy-image-dataset.

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
