# Peer review of "FedDL: personalized federated deep learning for enhanced detection and classification of diabetic retinopathy"

_PeerJ Computer Science, doi:10.7717/peerj-cs.2508_

## Round 0.1 · original submission · Major Revisions

Please, read carefully Reviewer #2 comments and address the comments to improve the submission.

·

Basic reporting

-Lines 54-68 Federal learning is moved to the Background section
- Lines 80-86 Authors mention section II, III, reflect this numbering in subsequent sections.
-Line 86 becomes II.Background and Related work, correct the other sections as well

Experimental design

Line 152: Proposed methodology: Authors must outline all necessary steps required to develop the FedDL model (e.g. you can use the CRISP_DM framework)
Line 332: In your study, the entire dataset is used for training only. Explain why you do not split the dataset into training, validation and testing.

Validity of the findings

Line 340: Performance metrics: Explain how you can use them in your dataset
Line 387: Change the name Round to Epoch in Table 3 and Table 4
Line 409: Explain why you chose the ResNet50 model for the new STARE dataset rather than your proposed model to validate the performance?

Additional comments

no comment

Reviewer 2 ·

Basic reporting

1. The introduction provides some background on Diabetic Retinopathy (DR) but lacks a clear statement of the motivation behind the study and its significance. Expand the introduction to better articulate the specific problem being addressed and why it is important. This could include discussing the current gaps in DR detection and classification and how federated learning can uniquely address these gaps. Providing quantitative data on the cases would also help.

2. The literature review does not fully explore relevant studies, and some references appear outdated or not directly related to federated learning in medical imaging. Include more recent and relevant studies on federated learning, particularly in the context of medical image analysis. Discuss how the proposed method compares to these studies and what new insights or advantages it offers.

3. Key terms and theoretical concepts such as “Federated Learning” and “Deep Learning Models” are not clearly defined, making it difficult for readers unfamiliar with these terms to understand the content. Provide clear definitions and explanations of all technical terms and concepts when first introduced. Consider including a brief subsection in the Introduction or a glossary for this purpose.

4. While the paper mentions the importance of privacy in federated learning, it does not discuss the ethical considerations in detail or the steps taken to ensure patient data privacy. Provide a more comprehensive discussion on ethical considerations, particularly around data privacy and security in the federated learning framework. Outline the specific measures taken to protect patient data in compliance with relevant guidelines or regulations.

Experimental design

1. The paper does not clearly outline the experimental design, such as the steps taken to preprocess data or the specific configurations of the models used. It would benefit from a detailed flowchart or diagram that maps the entire experimental process, from data collection to model evaluation, to enhance reproducibility and clarity.

2. The choice of models (ResNet50, DenseNet201, AlexNet, EfficientNetB7, and VGGNet19) is not adequately justified. The authors should provide a rationale for selecting these specific models, possibly comparing their performance with other state-of-the-art models in similar tasks to establish a stronger foundation for their selection.

3. The methodology does not explain the data preprocessing steps comprehensively. Including details on how the fundus images were prepared, what augmentation techniques were applied, and the reasoning behind these choices would provide better insights into the data preparation process.

4. There is no mention of a hyperparameter tuning strategy, which is crucial for optimizing model performance. The paper should include a section describing how hyperparameters were chosen, such as using grid search or random search, to ensure that the models are well-optimized.

Validity of the findings

1. The results section provides accuracy rates but does not contextualize these findings within the broader literature. Comparing these results with previous studies on diabetic retinopathy detection using similar methods would help readers understand the significance of the findings.

2. The use of accuracy as the primary metric is insufficient, especially in medical diagnostics where false positives and false negatives have different implications. The authors should include additional metrics, such as sensitivity, specificity, and AUC-ROC, to provide a more nuanced understanding of model performance.

3. There is no mention of overfitting or underfitting in the results. Including a discussion on these aspects, potentially supported by training and validation loss curves, would help understand the model's learning behavior and generalization capability.

4. Some graphs and tables lack detailed captions and explanations, making it difficult to interpret the data. Each figure should be accompanied by a comprehensive description to help readers understand the key takeaways.

5. While the conclusion mentions future work, the discussion does not elaborate on specific research questions or hypotheses that could be explored in future studies. A more detailed roadmap for future research would add depth to the discussion.

6. The discussion overly focuses on the positive outcomes of the study without adequately addressing the potential downsides or challenges encountered. A more balanced discussion that includes both strengths and weaknesses would provide a more objective assessment.

Additional comments

1. I have a lot of comments that needs to be answered before potential publication.

---

## Round 0.2 · Minor Revisions

We have received the reviewer’s comments regarding your manuscript. Below are the recommendations that, once addressed, will help improve the quality of your work:

Clarity and flow of sentences: While the reviewer has noted improvements in the findings section, they have requested further enhancement in the clarity and flow of sentences, including smoother transitions between sentences and paragraphs. We recommend carefully revising the text, aiming to simplify and clarify your ideas, and ensuring that each section flows naturally and cohesively.

Methods: Although the experimental design has been generally approved, the reviewer suggests providing more details in the methods section to promote reproducibility. Please include any additional information that might assist readers in accurately replicating your study.

We appreciate the effort you have put into improving your manuscript so far, and we are confident that with these adjustments, your work will reach an even higher level of quality.

Reviewer 2 ·

Basic reporting

1. Queries are answered reasonably.

Experimental design

1. Improved the details of the methods as to promote reproducibility.

Validity of the findings

1. Findings sections is now much better than before.

Additional comments

1. Please improve the clarity and flow of sentences, including smooth transitions from sentences and paragraphs.

---

## Round 0.3 · accepted · Accept

Congratulations on the acceptance of your manuscript.

Reviewer 2 ·

Basic reporting

Queries are answered.

Experimental design

The section is now fine.

Validity of the findings

Results and findings substantially improved.